# Orangutan Ecotourism on Sumatra Island: Current Conditions and a Call for Further Development

**Agus Purwoko** [1,*] , **Wanda Kuswanda** [2] , **Rospita Odorlina Pilianna Situmorang** [3] , **Freddy Jontara Hutapea** [2,4] , **Muhammad Hadi Saputra** [2] and **Parlin Hotmartua Putra Pasaribu** [2]

1. Faculty of Forestry, Universitas Sumatera Utara, Jl. Tri Darma Ujung 1 Padang Bulan, Medan 20155, Indonesia
2. Research Center for Ecology and Ethnobiology, National Research and Innovation Agency (BRIN), Gedung B.J. Habibie, Jl. M.H. Thamrin No. 8, Jakarta 10340, Indonesia
3. Research Center for Population, National Research and Innovation Agency (BRIN), Gd. Widya Graha Lt. X, Jl. Jend. Gatot Subroto No. 10, Jakarta 12710, Indonesia
4. School of Ecosystem and Forest Sciences, Faculty of Science, The University of Melbourne, Creswick, VIC 3363, Australia
* Correspondence: agus9@usu.ac.id; Tel.: +62-813-6051-3501

**Abstract:** This paper aimed to provide a narrative review of the current conditions of orangutan ecotourism on Sumatra Island, problems in the current management systems, and some recommendations for further development. Orangutan conservation centers have been developed on Sumatra Island since 1973. The Bukit Lawang Conservation Station is one of the orangutan conservation centers that have practiced ecotourism to overcome ecological and socio-economic issues. Even though the Bukit Lawang Conservation Station has operated for decades, this station has faced several issues, in particular a monetary crisis in 1997, a flash flood in 2003, and the COVID-19 pandemic. We identified that orangutan conservation centers on Sumatra Island have the potential to support orangutan ecotourism. These conservation centers have ecological support, available facilities, and rich local wisdom that can provide added value for orangutan ecotourism. Therefore, we propose that the development of orangutan ecotourism on Sumatra Island should accommodate surrounding communities through community-based wildlife ecotourism. We also recommend the following strategies to develop orangutan ecotourism on Sumatra Island: (1) mapping the location and distribution of wild orangutans in their natural habitats; (2) managing captive and semi-captive orangutans in conservation centers; (3) provision of tour packages; (4) community empowerment; (5) institutional strengthening of community-based ecotourism management (CBEM); (6) developing ecotourism through a benefit-sharing model; (7) anticipating and minimizing the negative impacts of ecotourism on orangutans; and (8) integrating orangutan tourism with local wisdom.

**Keywords:** Batang Toru; Bukit Lawang; community; national parks; tourism; wildlife

## 1. Introduction

Orangutans in Indonesia are classified as Bornean orangutan (*Pongo pygmaeus*), Sumatran orangutan (*Pongo abelii*), or Tapanuli orangutan (*Pongo tapanuliensis*) [1–3]. Orangutans are generally distributed in conservation forests (national parks, nature reserves, and wildlife reserves), protected and production forests managed under the Forest Management Unit (*Kesatuan Pengelolaan Hutan* or KPH), community lands, and company concession forests. Orangutans are considered a charismatic and unique species due to their physical characteristics (reddish fur and long facial hair) and other, behavioral characteristics, such as spending most of their time in the forest canopy and not living in family groups [4,5]. Sumatran orangutans and Tapanuli orangutans can only be found on Sumatra Island, while Bornean orangutans are only distributed in Borneo Island. Sumatran orangutans are mostly distributed in Gunung Leuser Natural Park (GLNP), Leuser Ecosystem, Trumon Singkil,

Rawa Tripa, Siranggas/Batu Ardan, and the Batang Toru Landscape. A few orangutans have been reintroduced into the Bukit Tigapuluh National Park (BTNP) and Jantho areas [6].

The orangutan population continues to decline due to deforestation, habitat degradation, and illegal hunting and trading [7–9]. To protect orangutans from extinction, numerous orangutan conservation centers have been established in Indonesia, especially on Sumatra Island. The most popular orangutan conservation center on Sumatra Island is Bukit Lawang Orangutan Rehabilitation Station, which was built in 1973 to rehabilitate orangutans translocated from conflict areas and those rescued from poaching and illegal trading [10]. The Bukit Lawang Orangutan Rehabilitation Station is managed collaboratively by local people and the central government. This conservation center is one of the most popular tourist destinations for local and international tourists. Between 1985 and 2009, the Bukit Lawang Orangutan Rehabilitation Station was visited by about 12,957 visitors annually [11]. Currently, the Bukit Lawang Orangutan Rehabilitation Station is called the Bukit Lawang Conservation Station [12].

Nowadays, the development of wildlife ecotourism is increasing due to people's growing interest in interacting with wildlife and the natural environment, especially in Asia and Africa [13]. The number of visits to several wildlife ecotourism facilities have increased remarkably. For example, the number of travelers to Semenggoh and Matang wildlife parks in Sarawak (Malaysia) increased by 67% from 1990 to 2011 [14]. The growth of wildlife ecotourism has increased the wildlife ecotourism market size. Newsome and Rodger [15] estimated that the global wildlife tourism market size is about USD 37 billion. The growth of wildlife ecotourism is also expected to bring economic benefits to the local economy. For instance, orangutan conservation in Semanggoh (Sarawak, Malaysia), which was visited by 70,000 visitors annually, contributed USD 23 million to the local economy [14].

Although orangutan conservation centers on Sumatra Island have existed for a long time, the management of such is still underdeveloped. Problems such as the pressure on orangutan freedom, high community dependence on external support, lack of collaboration with relevant stakeholders, inadequate facilities and infrastructures, conventional or outdated management, and the short-term orientation of management have impacted ecotourism development [16,17]. These conditions reflect a lack of willingness from the community to face the issues and a lack of government attention toward improving the development and sustainability of orangutan ecotourism on Sumatra Island. A recent study by Susilawati et al. [17] showed that the number of visitations to the Bukit Lawang Conservation Station between 2014 and 2016 decreased by 62%. Lockdown policies, travel restrictions, and the closure of wildlife ecotourism facilities during the COVID-19 pandemic are also expected to have a negative impact on ecotourism in Bukit Lawang. Until now, studies regarding the development of orangutan conservation for ecotourism on Sumatra Island have been limited.

This paper is a narrative review designed to summarize the literature, depict the current condition of orangutan tourism on Sumatra Island, and identify several problems in the current orangutan ecotourism management system. This paper also aims to outline supporting factors that have the potential to improve orangutan ecotourism on Sumatra Island. Moreover, this paper intends to formulate some strategies to strengthen orangutan ecotourism that have significant impacts on orangutan conservation programs and community empowerment on Sumatra Island. The reviewed materials used in this paper were obtained from relevant research papers, books, reports, and government policies. The information from these materials is described in detail. This paper is structured as follows: The next section describes the need to involve local communities in the orangutan conservation program on Sumatra Island. Section 3 presents government policies regarding the orangutan conservation programs and orangutan ecotourism and depicts the current condition of orangutan ecotourism on Sumatra Island. Section 4 identifies supporting factors to develop orangutan ecotourism on Sumatra Island. Section 5 provides some principles for managing orangutan ecotourism, and Section 6 presents the future development

of orangutan ecotourism management, followed by conclusions in Section 7. We expect this paper to provide helpful information and suggestions for the development of an orangutan conservation center on Sumatra Island.

## 2. The New Approach to Orangutan Conservation on Sumatra Island

The orangutan population on the Sumatra and Borneo Islands continues to decline [7–9]. The estimated Bornean orangutan population is about 57,350 individuals (29 metapopulations). The Sumatran orangutan population is around 13,710 individuals, distributed in their original habitats (eight forest units) and outside their original habitats (reintroduction). The Tapanuli orangutan population is about 577–760 individuals, distributed across three forest units [3,6,18,19]. To evaluate the sustainability of orangutans, the International Union for Conservation of Nature (IUCN) and the Orangutan Watcher Forums conducted a Population and Habitat Viability Assessment (PHVA). Through this activity, it was reported that most of the orangutan metapopulations in Indonesia are less than 200 individuals, far below the minimum requirement to be viable in 500 years (500 individuals per metapopulation) [6].

The main causes of the decline in orangutan population are deforestation, habitat fragmentation, and illegal hunting [18,20,21]. The BPS, or Statistics Indonesia (*Badan Pusat Statistik*, BPS), reported that the forests on Sumatra Island were reduced by 12.62% between 2009 and 2020 [22,23]. Although the Ministry of Environment and Forestry of the Republic of Indonesia (MoEFRI) [24] has stated that the deforestation in Sumatra has tended to decline in the last two years, previous deforestations and habitat destructions have forced orangutans to move outside their natural habitats (community farmlands and agricultural plantations) and frequently trigger human–orangutan conflicts [25,26]. A conflict becomes more intense when orangutans raid valuable crops such as king fruit (*Durio zibethinus* Murray), stink bean (*Parkia speciosa* Hassk), and jering (*Archidendron jeringa* Jack) [5]. Frequent crop-raiding incidents have caused local people to have a negative perception of orangutans. Most surrounding communities perceive orangutans as pests, and thereby they attempt to deter orangutans from entering their farmlands and even kill them to protect their crops [27–29].

On the global scale, the IUCN also categorized Sumatran orangutans, Borneo orangutans, and Tapanuli orangutans as critically endangered species in 2001, 2016, and 2017, respectively [7–9]. To prevent the decline of the orangutan population, the Government of Indonesia has categorized orangutans as a protected species (the Minister of Environment and Forestry Regulation No. 106/2018). In 2019, the Government of Indonesia published a conservation strategy and action plan (*Strategi dan Rencana Aksi Konservasi* or SRAK) for orangutans 2019–2029, but it has been withdrawn for further revision [30]. This SRAK consists of an in situ and ex situ orangutan conservation plan, institutional capacity development, and improvement of community awareness.

To improve orangutan conservation in the future, it is important to manage forests through the community [5,31]. Community-based forest management (CBFM) is a sustainable forest management model designed to provide various benefits, including ecological and socio-economic benefits [32]. Through this model, the government provides access to surrounding communities to allow them to be actively involved in forest management and gain some of the benefits to improve their livelihoods. This concept has been widely applied around the world. This model has also been adopted in wildlife conservation through community-based wildlife ecotourism. Through this approach, surrounding communities are allowed to participate in wildlife conservation programs and experience some of the benefits of these programs, including orangutan habitats [33,34]. Community-based wildlife ecotourism is also expected to solve human–wildlife conflicts and increase community participation in independent conservation [35,36].

### 3. Current Orangutan Ecotourism in Sumatera and Related Problems

*3.1. Government Policies to Support Orangutan Conservation and Ecotourism*

The Government of Indonesia has issued several regulations to protect and utilize the flora and fauna. These regulations consist of the Conservation of Biological Natural Resources and Ecosystem Law No. 5/1990 [37], the Forestry Law No. 41/1999 [38], and the Government Regulation No. 8/1999 [39]. The government also published the Regulation of the Minister of Forestry No. 447/2003 concerning the administration directive of harvest or capture and distribution of the specimens of wild plants and animal species [40]. To support nature-based tourism in forests and sanctuaries, the government released Government Regulation No. 36/2010 and the Minister of Environment and Forestry Regulation No. P.13/2020.

To support community-based ecotourism, the government has launched the Social Forestry programs through Government Regulation No. 6/2007 and the Minister of Environment and Forestry Regulation No. 9/2021 [41]. To improve the benefits of forests to state revenue, the government has established the KPH through the Minister of Forestry Regulation No. P. 6/2009. The KPH is a forest management area in protected and production forests that governs forestry business and develops collaborations with other parties to produce alternative schemes [42,43]. Wildlife ecotourism is one of the goals of KPHs inhabited by Sumatran and Tapanuli orangutans. Even though wildlife ecotourism has been practiced in numerous wildlife conservation centers in Indonesia, a technical regulation concerning wildlife ecotourism management is not available yet.

*3.2. Existing Condition of Orangutan Ecotourism on Sumatra Island and Related Problems*

The Bukit Lawang Conservation Station is the only orangutan conservation center on Sumatra Island that has successfully synergized conservation and ecotourism, while other orangutan conservation centers (in Jantho, Batu Mbelin, and Sungai Pengian) are still not fully implemented forms of ecotourism, although they have the potential for it. The Bukit Lawang Conservation Station is located in Bahorok, Langkat District, North Sumatra. This conservation center is located about 80 km from Medan (the capital city of North Sumatra Province) (Figure 1). The Bukit Lawang Conservation Center was founded by the Netherlands Foundation for the Advancement of Tropical Research (WOTRO) to rehabilitate orangutans rescued from conflict areas and illegal hunting [10]. In the early period, the Bukit Lawang Conservation Station attracted international visitors who were interested in wildlife conservation and tourism. In the following years, this conservation center successfully attracted domestic visitors [44]. From 2001 to 2014, there were around 8400 international visitors per year (around 700 people per month) and 40,443 domestic visitors [17].

The Bukit Lawang Conservation Station has been run to conserve orangutans since 1973. As in Sabah, Malaysia and Borneo Island, Indonesia, the purpose of orangutan conservation development is to rehabilitate orphaned, wounded, and abandoned orangutans to rebuild their strength and allow them to return to the forest [45–47]. In the Bukit Lawang Conservation Station, visitors are allowed to have interactions with captive and semi-captive orangutans, such as feeding, touching, and taking pictures. Visitors can also observe wild orangutans in their natural habitats. Bukit Lawang also possesses other natural beauty spots, e.g., the tropical rainforest, rivers, caves, and beautiful topography that may attract tourists.

The Bukit Lawang Conservation Station is managed through a co-management model between local people (Bukit Lawang villagers) near the Bahorok River and the GLNP Agency (central government). The government monitors orangutans and their habitats and maintains all facilities in the conservation center. Local communities, on the other hand, manage tourism activities by providing tour packages, attractions, and tour guides and developing supporting facilities around the conservation station. Through this collaboration, the government and local communities benefit from tourism and conservation

activities. Unlike the orangutan rehabilitation center in Sipilok, Sabah, Malaysia, ecotourism management involves few local people and local ecotourism business is minimal [45].

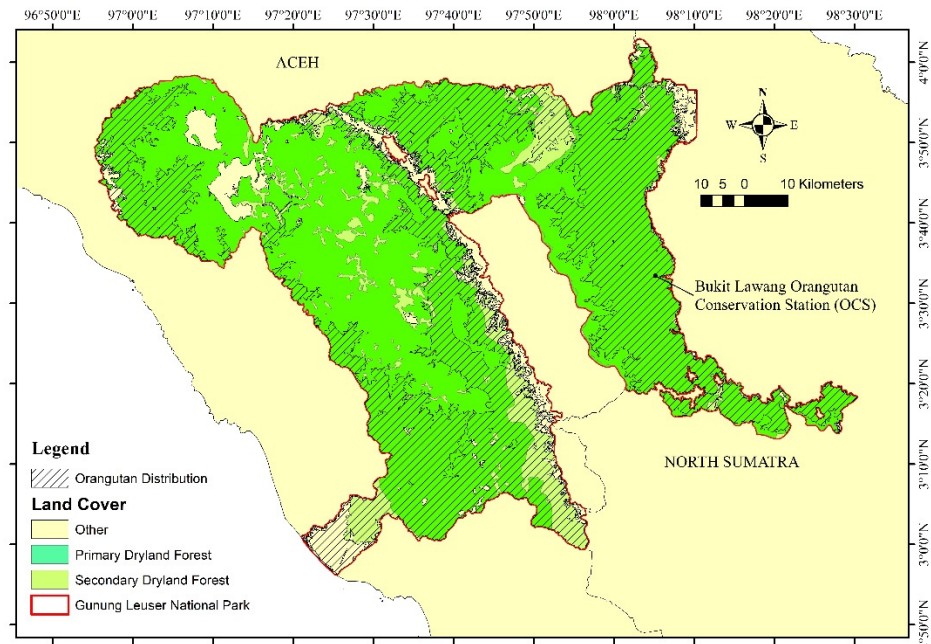

**Figure 1.** Distribution of orangutan habitats in GLNP with land cover of primary and secondary dryland forests [12].

However, the Bukit Lawang Conservation Station has faced numerous challenges in the last 25 years. The monetary crisis in 1997, the flash flood of Bahorok River in 2003, and regional autonomy in 2009 caused a decrease in the number of visitors to this location [17]. The lockdown policies, travel restrictions, and the closure of the conservation center during the COVID-19 pandemic have also had a negative impact.

From an ecological perspective, the GLNP is highly threatened by forest destruction, particularly from neighboring communities, palm oil plantations, and stone mining [17]. From a law enforcement point of view, the prosecution of illegal logging activities in GLNP is still poor. From an ecotourism management perspective, negative tourism activities, e.g., tourism encroachment into the national park, are occasionally observed in Bukit Lawang [48,49]. Compared to the Wehea-Kelay landscape, one of the natural habitats of orangutans in East Borneo, Indonesia, the government has established an essential ecosystem and corridor for orangutans. The designation of essential orangutan habitats involves various stakeholders including central and local governments, private sectors, and communities to improve and harmonize programs in relation to orangutans in the Wehea-Kelay landscape [50].

Bukit Lawang also has a conventional management system, poor access, and poor waste management that may reduce the number of visitors [16,17]. As with orangutan conservation and the ecotourism center in Sebangau Natural Park, East Borneo, Indonesia, poor access and a lack of facility development are the main problems for promoting ecotourism [51]. This evidence clearly shows that improvements are required to optimize ecotourism in orangutan conservation centers.

Compared to other regions, especially after the above-mentioned events, orangutan ecotourism on Sumatra Island has fallen behind Kalimantan Island (Tanjung Putting National Park—TPNP). In the late 1990s, Rijksen and Meijaard [52] estimated that the annual revenue of ecotourism in Tanjung Putting is about USD 240,000, while in Bahorok, it is only about USD 43,000–80,000. Past visits from celebrities, e.g., Julia Roberts and Bill Clinton, has also effectively promoted TPNP internationally [53]. TPNP has now been proposed as a priority tourist destination in Indonesia [54]. If compared with the orangutan

conservation center in Bukit Merah Island, Malaysia, the ecotourism management is better than in Bukit Lawang. In Bukit Merah, the ex situ Bornean orangutan ecotourism center is isolated on the island and is safer from human disturbances. The facilities and infrastructure to support ecotourism are advanced and interesting (transportation by boats) and managed effectively. Visitors expressed high satisfaction with their visit to the Bukit Merah orangutan conservation center, and were willing to pay up to twice the current ticket price [55].

## 4. Potential for Orangutan Ecotourism Development on Sumatra Island and Supporting Factors

### 4.1. Ecology

Most forests on Sumatra Island are tropical rainforests that are suitable for numerous flora and fauna. The GLNP, which covers North Sumatra and Aceh Provinces, is suitable for biodiversity conservation including orangutans [56]. The 830,000 ha GLNP is occupied by 536 identified plant species that can provide various types of food for orangutans [57]. The GLNP is also populated with abundant tall and compact trees and vines that are suitable for orangutans to climb, hang, play, and build their nests in. Moreover, the GLNP and Leuser Ecosystem are conservation forests.

Another forest inhabited by orangutans (*P. tapanuliensis*) is the Batang Toru Landscape. This landscape covers about 2750 km$^2$ of forests, where about 1383.4 km$^2$ of it is potential for orangutan habitat. The Batang Toru Landscape is administratively located in three regencies: North Tapanuli, Central Tapanuli, and South Tapanuli [58]. The Batang Toru Landscape comprises protected forests (51.5%), nature reserve areas (6.2%), production forests (5.3%), non-forested estates (36.8%), and water bodies (0.2%) [59]. The Batang Toru Landscape is occupied by 688 plant species, of which at least 191 species are food for orangutans [60]. According to Kuswanda et al. [61], the Batang Toru Landscape can maintain the population of orangutans when conflicts are minimized [61]. Moreover, the Batang Toru Landscape, especially on KPH (protected forests) and community land, has high biodiversity and unique ecosystems that have potential to be developed as tourist destinations [29].

The Bukit Tigapuluh National Park (BTNP) and Siranggas Wildlife Reserve are also suitable for orangutan habitats and orangutan ecotourism. Recently, the government released (reintroduced) more than 160 orangutans to BTNP (outside their original habitats) [62]. The orangutan-reintroduction program aims to give confiscated orangutans a second chance to live and reproduce in the wild without human intervention [63].

### 4.2. Supporting Facilities

To support orangutan conservation programs, numerous conservation centers have been developed on Sumatra Island, including the Bukit Lawang Conservation Station (North Sumatra), Orangutan Rehabilitation Center in Jantho (Aceh), Batu Mbelin Orangutan Quarantine Station (North Sumatra), and Sungai Pengian Orangutan Rehabilitation Station in Jambi (Figure 2) [64]. Numerous shelters and research stations have been built to rehabilitate problematic orangutans and monitor their conditions [10]. These conservation centers also perform domestication, breeding, behavioral and health monitoring, routine feeding, and training for the officers, which can potentially be synchronized with the ecotourism model [10]. Some research centers, such as Bukit Lawang, already have lodging facilities (hotels, inns, homestays) and vendors for selling food and transportation services [16,65].

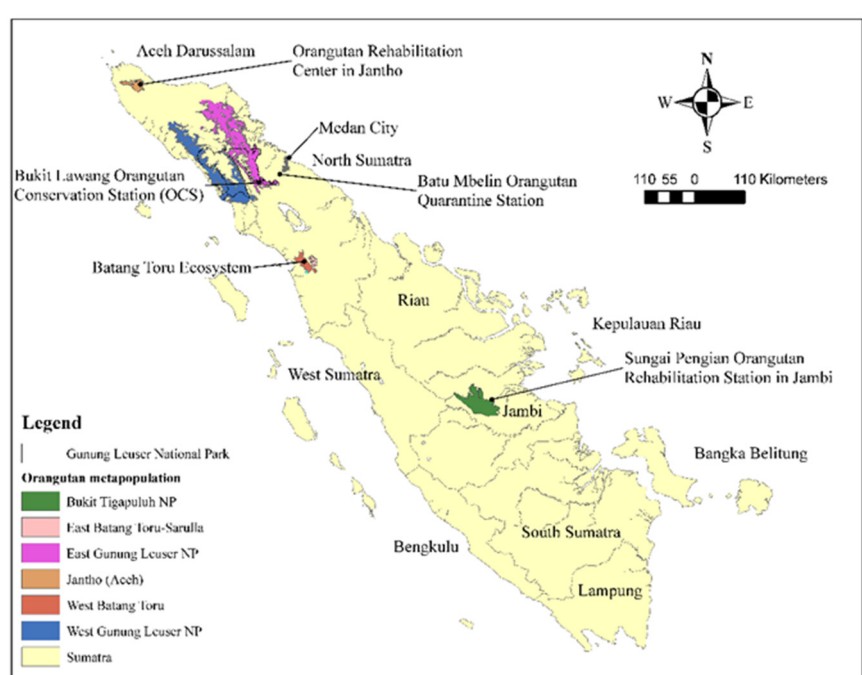

**Figure 2.** Orangutan metapopulations on Sumatra Island. The map was created based on the information in the Statistics of Forestry 2020 and the Convention on Biological Diversity 2014 [64].

### 4.3. Community Participation and Local Wisdom

Local community participation is one of the basic principles in developing community-based ecotourism. In the Bukit Lawang Conservation Station, the participation of local people in orangutan ecotourism is high because they are highly dependent on forest resources and ecotourism has benefited communities economically and socially [16,66]. The profit gained from orangutan tourism in Bukit Lawang was IDR 1,721,082,350 in 2018, and there are many other sources of income from this conservation tourism such as parking fees, restaurants, stalls, hotels, homestays, photos, and rental fees for tourist facilities [67]. By taking advantage of the economic and social benefits of orangutan ecotourism, local communities are simultaneously showing that they are willing to protect the forests from destruction (illegal logging and illegal mining) [35].

In general, the surrounding communities on Sumatra Island are rich in local wisdom [29]. In Bukit Lawang and Batang Toru, local communities traditionally protect water sources, watersheds, and forbidden forests which are also orangutan habitats. Local communities in these regions have also not cut down certain tree species, such as the *beringin* tree (*Ficus benjamina* Linn) which is believed to be a living house of ghosts. The fruits of *beringin* are food for orangutans and other animals [68]. In Tapanuli, local people also traditionally harvest the sap of the *aren* (*Arenga pinnata*) palm tree (*maragat*) and perform rituals before harvesting kemenyan (*Styrax spp.*) sap (*marhottas*). Moreover, the Karo tribe-dominated villagers in Bukit Lawang also have traditional medicines [69]. This evidence clearly shows that surrounding communities have much local wisdom that can provide added value for orangutan ecotourism.

### 4.4. Institutional Support

The Bukit Lawang community is the main actor in orangutan ecotourism management, while the GLNP officers are responsible for conservation activities. To conduct ecotourism governance within communities and with external parties, the Bukit Lawang community has institutional supports, such as government regulations, community organization, co-management and cooperation documents, and internal rules. The supporting regulations in developing community-based ecotourism in Bukit Lawang consist of Government Regu-

lation No. 36/2010 concerning nature tourism business in wildlife sanctuaries, national parks, grand forest parks, and nature tourism parks; Director-General of KSDAE Regulation No. SK. 193/2019 concerning the GLNP zoning area that designates the utilization zone of GLNP for multiple purposes including ecotourism; and Government Regulation No. 12/2014 concerning the type and tariff of the state's non-tax revenues in the Ministry of Forestry. The institutional support available in Bukit Lawang orangutan ecotourism is exemplary and can be implemented in other orangutan conservation centers if they are managed through ecotourism.

The Bukit Lawang community has also collaborated with many institutions to develop and support orangutan ecotourism. These collaborations take the form of ecological provision, technical and management support, capacity development, facility development, infrastructure development, and promotional support (Table 1). The community has also provided tour packages and trained 200 tour guides to improve ecotourism attractions and services in Bukit Lawang [16,17].

**Table 1.** Types of collaboration between the Bukit Lawang community and other parties.

| No. | Type of Collaboration | Collaborative Parties | References |
|---|---|---|---|
| 1 | Ecological provision of GLNP utilization zone | GLNP Agency/*Balai Besar* | Syahputra [16]; Susilawati et al. [17]; Siburian [65] |
| 2 | Ecological conservation | MoEFRI, Yayasan Ekosistem Lestari (YEL) | Sumatran Orangutan Conservation Program, SOCP [70] |
| 3 | Technical and management support | GLNP Agency, MoEFRI, WWF | Rijksen [10]; Syahputra [16]; Susilawati et al. [17] |
| 4 | Capacity development | MoEFRI, NGOs, universities | Syahputra [16]; Susilawati et al. [17]; Siburian [65] |
| 5 | Facility development | WOTRO, WWF, MoEFRI | Rijksen [10]; Siburian [65] |
| 6 | Infrastructure development | Local government | Syahputra [16]; Susilawati et al. [17]; Siburian [65] |
| 7 | Promotional supports | Travel agencies, local and central governments, NGOs | Syahputra [16]; Susilawati et al. [17] |

*4.5. Ecosystem and Natural Uniqueness*

The Bukit Lawang Conservation Station in GLNP and Baharok River is the main spot that may attract domestic and foreign tourists. This spot has unique characteristics. GLNP is high in biodiversity, while the Bahorok River has a strong stream that makes it suitable for river-based tourism (rafting, swimming, and tubing) (Figure 3) [71]. GLNP is the only national park in the world that has orangutans, elephants, tigers, and rhinos, which are endemic to Sumatra [72]. YOSL-OIC [73] pointed out that these animals have the potential to attract many parties for conservation, education, research, adventure, and tourism activities. GLNP also varies in altitude (30–1200 m) and has various slopes (dominated by high slopes) that are suitable for paragliding. Moreover, the steep topography and small rivers flowing into the Bahorok River create several waterfalls in GLNP. Trekking and camping can also be used for tourism in the GLNP. In addition, Bahorok has karst caves connected to rivers that can be used for river tubing and cave exploration.

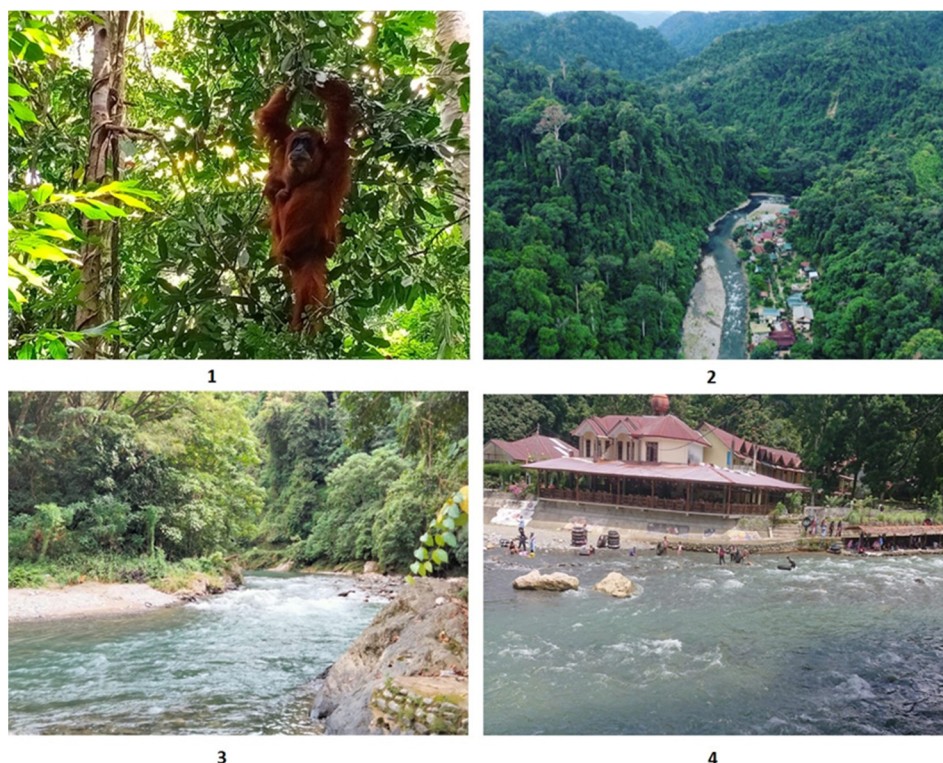

**Figure 3.** (**1**) Sumatran orangutan (*P. abelii*); (**2**) forest ecosystem; (**3**) river for rafting; and (**4**) homestay in the Bukit Lawang Conservation Station (photo source: GLNP Agency).

## 5. Principles of Orangutan Ecotourism Management

### 5.1. Ensuring Orangutan Welfare

Ecotourism practices in conservation centers should ensure animal welfare by providing feasible nutrition, ensuring a natural and friendly environment, applying natural behavioral interactions between humans and animals, monitoring health conditions, and ensuring a stable mental state [74]. The development of endangered wildlife ecotourism should also extend the expected survival time of wildlife [75].

Orangutans need wide areas to live in with numerous big trees. The daily roam area is around 0.7–26.2 ha/day [76]. Generally, orangutans live solitarily, except during the breeding season and when raising their young [5]. Most orangutans' daily activities in their natural habitats, such as swinging from one branch to another branch, climbing, and hanging, are highly dependent on trees. Orangutans also build a new nest every day near a food source and quite high above the ground [61,76–78]. The abundance of feed is a necessity for orangutans as they consume around 6.2 kg of food/day/individual [79]. Considering their daily needs and their behaviors, maintaining primary and secondary forests for their habitat is vital to support their survival.

### 5.2. Empowerment and Economic Improvement of Communities

The involvement of local communities in ecotourism, either full or partial involvement, is important to develop these communities socially and economically. Local people can be employed as research assistants, staff, cleaners, drivers, and tour guides at an orangutan conservation center. Facilitations and training programs are required to improve community knowledge and skills. Ecotourism may also promote local wisdom, such as traditional culture, cuisine, and traditional medicines that may provide more job opportunities to local communities [80,81]. Overall, ecotourism will generate multiplier effects on communities and the region.

In recent years, there has been growing interest in using the surrounding community as the subject of conservation programs. In this study, we also propose that local people should

be the subject of orangutan ecotourism on Sumatra Island (community-based ecotourism). Through this concept, the well-being of the local community near the orangutan habitat can be improved by providing jobs for local people. This concept will also improve the responsibility of local communities to protect wildlife habitats and to succeed in wildlife conservation programs.

*5.3. Education and Visitor Satisfaction*

One of the goals of orangutan ecotourism is providing education for visitors. Visitors can learn about orangutans' behavior, habitat, and conservation practices [82,83]. To support this learning process, the management needs to provide information about orangutans and their conservation programs (audio, printed, and digital media), facilities, and tour guides [84]. Ecotourism should also improve the visitors' awareness of wildlife conservation [66,85].

Visitor satisfaction is an indicator of how far ecotourism fulfills visitors' expectations. It is generally shown by visitors' appreciation of the trip and their willingness to share details of it with others [86]. According to Newsome et al. [87], visitor satisfaction has a great role in ensuring the long-term sustainability of wildlife ecotourism. To increase the number of visitors, orangutan ecotourism needs to maintain visitor satisfaction by improving the safety and comfort of tourism activities, and providing proper facilities, services, information, and accessibility [88].

*5.4. Opportunities for Orangutans and Humans to Coexist through Ecotourism Management*

Harmonious coexistence is the main goal of human–wildlife conflict mitigation programs [36]. According to Carter and Linnell [89], harmonious coexistence between humans and orangutans can be achieved if humans and orangutans can co-adapt to living in shared spaces where human interactions with orangutans are governed by effective institutions that ensure long-term orangutan population persistence, social legitimacy, and tolerable levels of risk. Ecotourism is one of the main solutions to create harmonious coexistence between humans and wildlife, especially orangutans. Through ecotourism, local people are expected to receive economic benefits from tourism activities that ensure that they do not rely heavily on forest resources [36]. Simultaneously, local people are expected to have positive perceptions of wildlife and be actively involved in wildlife conservation programs.

Jelfi [90] has provided evidence that ecotourism altered local people's perceptions of nature and orangutans in Sebangau National Park, Kalimantan. In the beginning, local people refused the establishment of Sebangau National Park as they relied heavily on forests. Programs designed to involve local people in conservation programs, especially orangutan conservation programs, and the development of ecotourism in this national park improved the economic life of local people and altered their attitudes toward nature and orangutans.

## 6. Future Development of Orangutan Ecotourism Management

Based on our identification and review of the current conditions of orangutan ecotourism on Sumatra Island, we found that the development of orangutan ecotourism centers on Sumatra Island still needs improvement. In the studies that we reviewed, we found a gap between Bukit Lawang and other conservation centers in Sumatra compared to other advanced conservation centers such as in Malaysia and Borneo, particularly in management, technology, financial support, and protection of orangutans and natural resources.

The future management of orangutan ecotourism on Sumatra Island should emphasize local community empowerment because local people in Sumatra still depend on forest resources for their livelihood. Generating benefits from ecotourism can reduce forest destruction such as illegal logging, conflicts, and encroachment. Ross and Wall [91] pointed out that the foundations for successful ecotourism are the protection of natural forests, the establishment of high-quality tourism experiences, and local economic growth. In

developing orangutan ecotourism, the dynamics between orangutans, local people, and ecotourism that bring positive contributions to others need to be maintained (Figure 4) [92].

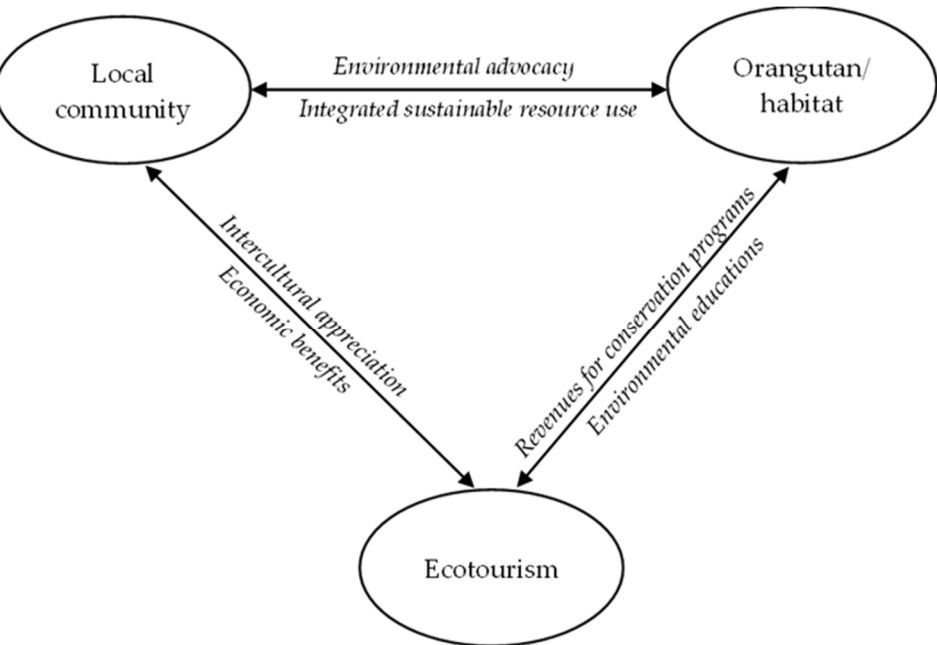

**Figure 4.** Orangutan ecotourism paradigm adapted from Ross and Wall [92].

The future management of orangutan ecotourism should also consider the different conditions and needs of orangutans in the conservation centers. Orangutans in the conservation center are categorized as wild, captive, or semi-captive; therefore, ecotourism management needs to treat them accordingly. Proper management aims to reduce the negative impacts of tourism on orangutan behavior change and conservation [66]. Therefore, management should ensure the fulfillment of orangutan welfare, provide visiting guidance to educate visitors, and provide for the safety of orangutan tourism.

Considering the existing management, the basic principles of ecotourism, the potential, and the problems, we recommend various strategies to increase the successful management of orangutan ecotourism management on Sumatra Island, set out in the following subsections.

### 6.1. Mapping the Location and Distribution of Wild Orangutans in Natural Habitats

Knowing the location and distribution of wild orangutans is essential for successful wildlife ecotourism. Wild orangutans are itinerant with a wide home range. They also build their nest near a source of food depending on the season. Mapping the location and the distribution of wild orangutans in their natural habitats can be used as a foundation to predict the presence of orangutans in certain areas in different seasons, thereby improving the chance of encountering orangutans in their natural habitats [93]. To produce this map, management needs information on orangutan distributions, types of forest, zonation, topography, source of food potency, and seasons. In Batang Toru Landscape, we also recommend mapping the distribution of orangutans in the KPH IX as this area has a large orangutan population.

### 6.2. Managing Captive and Semi-Captive Orangutans in Conservation Centers

The expected practice of the orangutan conservation program is to release orangutans into forests and protect their natural habitats. However, high pressure on their habitats, conflicts, poaching, and smuggling have encouraged the development of orangutan conservation centers to rescue and rehabilitate problematic orangutans and solve financial problems of wildlife conservation [34,55,94]. In orangutan conservation centers, the

rescued orangutans are domesticated and have become captive or semi-captive. Some orangutan conservation centers in Malaysia and Indonesia have established semi-natural habitats [55,95,96]. In this system, orangutans are housed from the evening to the next morning and then released to let them experience their natural habitats. Orangutans can also be free to choose where to sleep in their surrounding areas [10]. The release of orangutans also allows them to gradually return to their wild behavior and socialize with other members or groups.

The habitat of orangutans should be enriched with edible food to accelerate the adaptation of orangutans to the wild habitat and make them more independent. To ensure the fulfillment of their nutritional requirements, the diet consists of cooked rice, bananas, papaya, cabbage, carrots, maize, beans, tomatoes, cucumber, milk (for juveniles and adolescents), and many other foods that can be given routinely [10]. Sick animals should be separated and cured.

Orangutans like hanging, climbing, and playing. Therefore, the development and architecture of orangutans' physical environment in conservation centers need to accommodate these characteristics [95].

### 6.3. Provision of Tour Packages

Tour packages aim to provide information about ecotourism, promote ecotourism, and provide several tourism options for visitors. In developing tour packages, management should consider the condition of orangutans (wild or captive, aggressive or not) and the safety of visitors. For wild orangutans, a tour package can take the form of observing orangutan activities in their natural habitats (climbing, nesting, nurturing their babies, and playing) [16,96,97]. For captive and semi-captive orangutans, visitors can feed them, make physical contact, and take pictures with them. These tour packages offer a face-to-face experience that is usually interesting to visitors [16,97]. The packages can also be developed into a day visit or more-than-a-day visit, depending on the purpose of the visit. A day visit is set to meet and interact with orangutans, while several days' visit is set for research and documentary activities. For advanced conservation centers, safari rides with jeeps and motorcycles could also be developed.

Tour packages can also be mixed with other potential tourist attractions, e.g., jungle trekking, camping, rafting, paragliding, river tubing, and cave exploration, to improve visitor satisfaction. The tour packages can also be mixed with tourism of local customs, such as culinary traditions, traditional lodges, local herbal traditions, and traditional attractions, to introduce local customs to visitors. Promotion of tour packages can be conducted through advertisement services (television, radio, and newspaper), websites and social media (Facebook, Instagram, and TikTok), and public relations methods [98].

### 6.4. Community Empowerment to Improve Community Welfare

Community empowerment can be conducted through the involvement of communities in numerous programs. Local people are educated about the important roles of orangutans in the ecosystem, and then trained to produce alternative livelihoods such as those involving souvenirs and locally made foods that can bring economic benefits to the local community. For example, in Sintang (West Kalimantan), the women in the orangutan habitat are trained to produce orangutan dolls that can be sold to visitors [99]. Meanwhile, in Sebangau National Park, the women are trained to produce locally made food from fish, e.g., crackers and shredded fish, and the men are prepared to act as tour guides [90].

Moreover, community welfare can be improved by preparing the community to run small tourism enterprises, e.g., lodging, transportation, and environmentally friendly restaurants. These activities will simultaneously provide jobs for local people near the orangutan conservation centers [13]. However, the establishment of small tourism enterprises requires large investments. This problem can be solved through the development of community enterprises, a joint venture with private investors, or through local government programs.

### 6.5. Institutional Strengthening of Community-Based Ecotourism Management (CBEM)

Institutions are fundamental in providing internal and external governance of community-based forest management [100]. Internal governance reflects community resilience in managing their resources and overcoming problems and potential conflicts [35]. External governance reflects the community's relations with other parties and efforts to improve profitable collaborations with other parties [80].

Institutional strengthening of CBEM in orangutan ecotourism can be conducted internally through the establishment of the community organization or community organization deeds. This organization should have a robust organizational structure, purposes and missions, rules and norms, and community members and responsibilities. The community capacity also needs to be improved through knowledge sharing, workshops, and training.

The general problem of CBEM, including the Bukit Lawang Conservation Station, is its financial dependency on the founder institutions (government and international NGOs) [17]. Another problem of CBEM is the lack of short-term cooperation, and the absence of coherent joint planning with important or relevant stakeholders [34]. Moreover, the technology used to operate CBEM and promote orangutan tourism is outdated and ineffective, particularly in the digital era. Furthermore, accessing orangutan ecotourism is generally difficult, as wildlife conservation centers are usually located in remote areas [16].

To overcome these issues, the communities need to improve their organizational management by wisely sharing their profits for community development and saving their profits for operational management. Communities also need to improve their collaboration with relevant stakeholders in operating ecotourism (Figure 5). The institutional model for community-based collaborative ecotourism management in Tangkahan, an approach to wildlife tourism in GLNP, can be a role model for CBEM [101]. The collaboration model can be developed in the form of co-managements, cost and profit-sharing, and raising grants. The tourism sector also needs to update its technologies and provide interesting and updated information on websites. Promotional materials for tour and travel agencies, research institutes, and universities can be improved through collaboration. Strengthening the collaboration with government institutions is also very important to support facility and infrastructure development, as well as law enforcement against forest destruction. Strengthening collaboration with other communities, traditional community leaders, and other CBEs around orangutan ecotourism is also important to protect the forest and develop the CBEM [65,102]. Collaboration with customary institutions and cross-village management has been practiced in Semende, South Sumatra, which proves the sustainability of forest conservation activities in the villages [103].

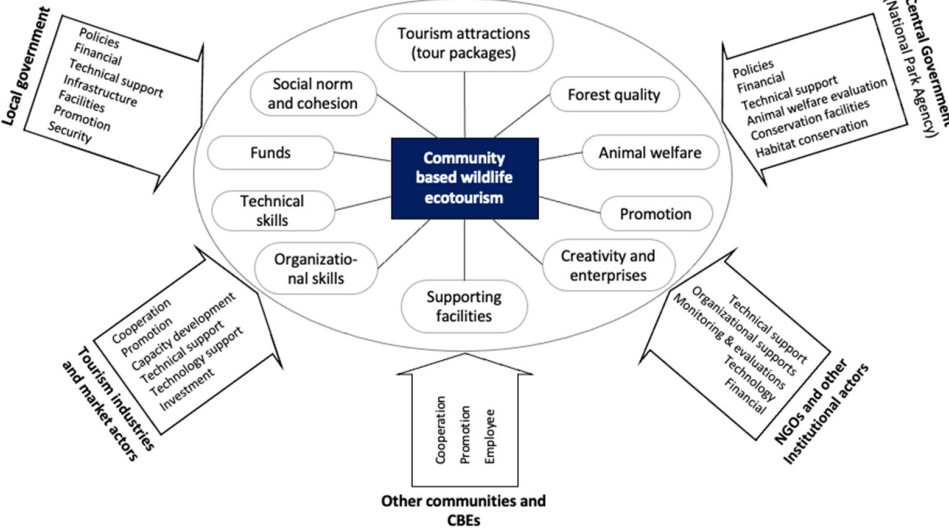

**Figure 5.** Institutional strengthening of community-based ecotourism management.

### 6.6. Developing Ecotourism through a Benefit-Sharing Model

One of the challenges for managing community-based conservation and ecotourism is to maintain its sustainability. Numerous community-based projects have failed because they are more focused on administrative procedures, tend to serve elite interests, and do not provide significant benefits to other parties [104–106].

From a benefit-sharing perspective, orangutan ecotourism should also provide fair benefits to collaborative parties. For example, collaboration with government institutions should benefit the government ecologically and economically. Collaboration with private sectors and tour agencies should provide economic benefits to both parties [107], while collaboration with education and research institutions is expected to bring economic and social benefits (Table 2).

**Table 2.** Offered benefits to collaborative parties.

| No. | Collaborative Parties | Offered Benefits | | |
| --- | --- | --- | --- | --- |
| | | Ecological Benefits | Economic Benefits | Social Benefits |
| 1 | Government institutions (central government and local government) | Maintaining forest quality, improving forest security, conserving wildlife | Tax revenue from entry tickets and ecotourism facilities (hotels, restaurants, etc.) | Reducing conflicts |
| 2 | Private sectors (industries, markets, and hotels) | Quality of the environment, business sustainability | Investment, profits sharing | Promotion (business image) |
| 3 | Tour agencies | Business sustainability | Discount price, profits sharing | Promotion (business image) |
| 4 | Education and research institutions | Ecological management model | Price discount | Learning experience, research site |

Mutually beneficial collaboration can improve visits to ecotourism areas [14]. In the longer term, further development (investment) in numerous sectors will be available, and local communities will obtain more benefits.

### 6.7. Anticipating and Minimizing the Negative Effects of Ecotourism on Orangutans

Negative externalities often occur in the development of wildlife conservation through ecotourism due to its inability to ensure the long-term protection of environmental assets. Shannon et al. [108] highlight the potential environmental impacts of ecotourism development, such as habitat degradation due to the establishment of infrastructure, human waste and litter, and chemical, light, and noise pollution. It may happen when the managers ignore or miscalculate the external cost of the damage to living resources [109]. Managers tend to increase income by over-exploitation, and neglect of animal welfare rules may happen due to the lack of commitment, understanding, and supervision. If these things are ignored, it is prone to market failure.

Anticipation and minimization of the negative impacts of orangutan ecotourism could be achieved by limiting attractions and interactions between visitors and orangutans that can cause orangutans stress and change their natural behavior [11,110]. Animal welfare fulfillment needs to be enforced, and the setting of the number of visitors per day or visit-group can anticipate over-exploitation. If the number of visits is due to increase, increasing ticket prices can be considered so that conservation and economic objectives can be achieved. Externality costs-and-returns for conservation activities should be accurately calculated to achieve marginal benefits. Educating managers, tour guides, and local people is important to improve their awareness and capability [11]. The managers should also

effectively educate the visitors by providing tour guidelines (oral guidelines, leaflets, and warning/caution boards) and trash containers in the areas to improve visitors' awareness. Implementing visitors' volunteer work in conservation centers as practiced in Samboja, Borneo, Indonesia, is quite effective in improving visitors' knowledge, skill, and awareness and promoting sustainable practices [47]. The establishment of supporting facilities around orangutan ecotourism also needs to consider the environment by using environmentally friendly materials. Infrastructure such as jungle tracking, observation points, security checkpoints, and survey land cameras need to be provided and developed to improve visitor safety and reduce human disturbances of natural resources and wildlife. The government, as the legitimate provider, needs to administer and provide clear regulations and technical guides to ecotourism managers [109].

*6.8. Integrating Orangutan Tourism with Local Wisdom*

Local wisdom has the potential to provide added value for orangutan conservation centers. To successfully support orangutan ecotourism, the community needs to improve the marketing of their local wisdom. Andari et al. [111] states that traditional events offered to tourists should be special, enjoyable, and memorable. Andari et al. [111] have suggested that local communities also need to improve their hospitality to welcome visitors.

Integration of wildlife-based tourism management with local knowledge and local rules is important to improve social capital. It can improve community resilience and ecotourism sustainability as practiced by the Muara Baimbai Community in Sei Nagalawan Village, North Sumatra [35]. It can improve motivation, sense of belonging, and responsibility for conservation activities in the ecotourism areas that affect sustainable forest management. The expected local wisdom here is mainly related to the interaction between humans and forests, both as an ecosystem and as a habitat for orangutans. Local wisdom can be packaged into tourist attractions that support orangutan conservation. Participatory research is needed to explore the models of integration between local wisdom and forest conservation activities in various natural habitats of orangutans. In the future, a model of sustainable human–animal co-existence can be developed to promote education-based tourist attractions.

## 7. Conclusions

The orangutan population on Sumatra and Borneo Islands continues to decline due to deforestation, habitat degradation, poaching, human–orangutan conflicts, and illegal trading. To protect orangutans from extinction, the government of Indonesia has classified orangutans as a protected species. The orangutan conservation program on Sumatra Island was started in 1973, following the establishment of the Bukit Lawang Conservation Station. This station was developed for both orangutan conservation programs and ecotourism. This station has become a popular tourist destination on Sumatra Island. However, in the last 25 years, this station faced numerous issues such as a flash flood that destroyed the ecotourism facilities, low financial support, low enforcement of environmental issues, inadequate management by the local community, low facility developments, and the COVID-19 pandemic. These have resulted in a decline in visits to Bukit Lawang, particularly after the flash flood in 2003 and after the regional autonomous policy was implemented in 2014. This situation indicates that improvements are required to bolster ecotourism in orangutan conservation centers on Sumatra Island after the COVID-19 pandemic. We identified that numerous locations on Sumatra Island, as the natural habitat of orangutans, have the potential, in terms of uniqueness and natural beauty, to improve tourism activities around the orangutan conservation centers. Surrounding communities also have local wisdom that can be developed to provide added value to orangutan ecotourism. To develop orangutan ecotourism on Sumatra Island, we recommend that the management create a distribution map of orangutans in their natural habitats, manage captive and semi-captive orangutans in conservation centers, provide tour packages, empower the community,

strengthen the CBEM institutions, develop a benefit-sharing model, and integrate orangutan ecotourism with local wisdom.

**Author Contributions:** Conceptualization, methodology, analysis, validation, investigation, resources, writing, visualization, A.P., W.K., R.O.P.S., F.J.H., M.H.S. and P.H.P.P. All authors have contributed equally as main contributors. All authors have read and agreed to the published version of the manuscript.

**Funding:** This research received no external funding.

**Institutional Review Board Statement:** Not applicable.

**Informed Consent Statement:** Not applicable.

**Data Availability Statement:** Not applicable.

**Acknowledgments:** We acknowledge the Universitas Sumatra Utara, the National Research and Innovation Agency (BRIN), GLNP Agency, and all parties that have supported the manuscript.

**Conflicts of Interest:** The authors declare no conflict of interest.

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
