# Peer review of "Orangutan Ecotourism on Sumatra Island: Current Conditions and a Call for Further Development"

_sustainability, doi:10.3390/su141811328_

Round 1

Reviewer 1 Report

Obecnie rozwój ekoturystyki opartej na dzikiej przyrodzie wzrasta ze względu na rosnące zainteresowanie ludzi interakcją z dziką przyrodą i środowiskiem naturalnym, zwłaszcza w Azji i Afryce.

Artykuł jest ciekawy i cenny dla rozwoju ekoturystyki. Zawiera niezbędne praktyczne wskazówki.

Interesujący jest również przegląd literatury.

Artykuł powinien zostać opublikowany.

Author Response

Distinguished First Reviewer.

Thank you for your interest and the acceptance of the present draft (the first stage manuscript). However, we have added some information in the manuscript to accommodate the feedback from other reviewers. We hope the last version of this manuscript provides a much better presentation.

Reviewer 2 Report

The topic of the article is very interesting and falls within the topics agreed by the Tourism, Culture, and Heritage section. However, framing the article in the form of a review raises some issues of conceptualizing and reorganizing it.

In the next section, entitled "Current Orangutan Ecotourism in Sumatera and the problems", the authors present in particular the natural conditions in which orangutans live in Sumatra and government policies to support the conservation of this species, without presenting clearly and concretely the actual situation of ecotourism. in the area. We do not know a profile of the ecotourist in the area, an evolution in time of ecotourism phenomenon.

The following sections of the paper are the authors' proposals for the future organization and development of orangutan ecotourism on the island of Sumatra. The conclusions are brief and resume aspects of the introduction and abstract. We do not know what is new about the article, the added value brought to the scientific community.

In conclusion, if the authors wish to keep the article in the form of a review, they should reorganize the entire content according to a review of the literature on this topic. In my opinion, in this situation, the paper should answer the following questions: What is the current situation of the literature dedicated to the topic, what topics have been treated as a priority in previous papers? What is the gap in the literature identified by the authors and how can it be studied, capitalized? What are the important sub-topics of the topic that have research potential in the future and that have been analyzed less in previous papers?

If the authors want to resubmit the article in the form of a research, they can analyze the ecotourism in the island of Sumatra, but using more data that they will process according to the established objectives. One possibility would be to make comparisons between different areas where this form of ecotourism is taking shape, including the island of Sumatra.

Author Response

Distinguished Second Reviewer,

Thank you for your appreciation of this study topic and the valuable comments and suggestions for this manuscript. We have added some information to accommodate your comments and feedback from other reviewers. We construct this article in the form of a review (a narrative review) designed to summarize the literature, depict the current condition of orangutan tourism on Sumatra Island, and identify several problems in the current orangutan ecotourism management system (L93-96). Detailed information is in the following response.

Comment 1.

What is the current situation of the literature dedicated to the topic, and what topics have been treated as a priority in previous papers?

Response to Comment 1

Thank you for the valuable comment. We have presented and revised the information about the actual condition of orangutan ecotourism in Sections 3.2, and 4.2, and some information in the introduction to accommodate your comment.

In short, on Sumatra Island, it is only one orangutan center (Bukit Lawang Conservation Station) that has been run through ecotourism (The information is presented in L191-192). This area has been visited by domestic and international visitors (L203) with the number of visitors being 12,957 annually between 1985 and 2009 (L67). Visitors can interact with captive orangutans in the conservation center, such as feeding, touching, and taking pictures (L207-208). The conservation center is co-managed by the local people and government (L220-221). In the area, a number of facilities to observe and interact with orangutans and facilities to support visitors’ needs such as hotels, and transportation have been available (Subsection 4.2. L328-338). The area has provided social and economic benefits to the local community. However, the monetary crisis in Indonesia in 1997, a flash flood in 2003, pandemic Covid-19, and poor management have caused the diminishing of the tourism activities in Bukit Lawang Orangutan Conservation Station nowadays (L230-242). They all impact the local communities socially and economically. 

Our study puts priority to address the existing and the current problems of orangutan ecotourism on Sumatra Island (Section 3), the potency for development  (in Bukit Lawang and other rehabilitation centers on Sumatra Island) (section 4), and the standards (principles) of wildlife ecotourism for proper management of orangutan conservation center through ecotourism in the future (section 5). Therefore, based on the reviewed topics, this paper intended to formulate some strategies to strengthen orangutan ecotourism that have significant impacts on orangutan conservation programs and community empowerment on Sumatra Island (Section 6). This information has been provided in manuscript L93-111.

Comment 2

What is the gap in the literature identified by the authors and how can it be studied, and capitalized?

Response to comment 2

We have provided the gap in the present management of orangutan ecotourism in Sumatra Island that is compared to other orangutan ecotourism areas in Indonesia (Borneo Island) and In Malaysia (L244-268).  We identified that the development of orangutan ecotourism centres on Sumatra Island still needs improvement. The gaps are mostly in management, technology, financial support, and the activities to protect orangutans and natural resources, as presented in Section 6. This information is presented in L670-675.

Comment 3

What are the important sub-topics of the topic that have research potential in the future and that have been analyzed less in previous papers?

Response to Comment 3.

In this review article, we provide a formulation strategy for the better management of orangutan ecotourism in the future, based on our evaluation (comparing Sumatra and other areas and the current standards or principles), rather than suggest future studies of particular topics/issues that are still absent.

We have provided information about this review model in the introduction part (L94-96). This paper is a narrative review designed to summarize the literature, depict the current condition of orangutan tourism on Sumatra Island and identify several problems in the current orangutan ecotourism management system. Therefore we provide suggestions for the future development of orangutan ecotourism on Sumatra Island.

Reviewer 3 Report

The conservation of endangered orangutan habitats and the maintenance of the species is an important ecological objective, together with the systematic exploitation of the GDP-producing capacity of ecosystem services, in a way that maintains a balance between economic profitability and the protection and conservation of biodiversity.  The authors of this article have provided a fair analysis of the situation of orangutans on the island of Sumatra in terms of their habitats and populations and the status of ecotourism and related regulations. They have highlighted the problems associated with current ecotourism. Based on their analysis, they provided strategic suggestions for further development.

Their proposal is exemplary and could be adapted to the development of ecotourism in other endangered species' habitats where it is a significant GDP generating sector.

In addition to institutional strengthening of community-based ecotourism management, attention should also be given to how to minimise the negative externalities that can arise from increased ecotourism. Increasing tourism, travel and the number of visitors also has an environmental impact. How do you intend to address this? Through education, awareness raising, infrastructure development. This should be emphasised.

Author Response

Respected Third Reviewer,

Thank you for your comment and evaluation of the content and structure of our manuscript. In addition to institutional strengthening of community-based ecotourism management, we have revised our paper by adding one sub-section (6.7. Anticipating and minimizing the negative impacts of ecotourism on orangutans) in the attention to how to minimize the negative externalities that can arise from increased ecotourism, such as trough the cost-benefits valuation, emphasizing animal welfare, awareness raising, and infrastructure setting. Please kindly refer to L-945.

Reviewer 4 Report

The proposal about “Orangutan Ecotourism on Sumatra Island: Current Conditions and a Call for Further Development” is attractive. To improve, you need the next questions:

-Abstract: it is right, but you have to include a mention to methodology.

-Theoretical framework: it is correct. Check if some references can be updated. To be prudent, try to update some new references if you find.

-Method: the principal lack of this paper is Methodology. The question is that there is not method in a scientific strict sense. You must use a quantitative or qualitative method. I can guess a qualitative indirect method (more or less, a sort of case study messed with content analysis), but it is not clear. Although the main objective of the research is to purpose a prospective activity, a scientific paper needs a method, at least. In this framework, I suggest in-deep interviews or a Delphi.

-Results. The absence of method implicates a lack or results.

-Conclusion and discussion: You have to include a more in-depth discussion comparing your findings with the principal authors of the theoretical framework. Conclusion is too much brief. There is a clear disequilibrium between the article in general and conclusion.

Author Response

Distinguished Reviewer,

Thank you for the valuable comment on our manuscript. The paper about “Orangutan Ecotourism on Sumatra Island: Current Conditions and a Call for Further Development” is a review article. We need to inform that we did not do empirical research in the data collection, however, we collected some information and data from previous literatures (published papers/articles), identified the specific topics, and formulate the strategy for Sumatra Island based on the compared existing management of orangutan ecotourism in Sumatra and other places and general principles.

We have tried to revise our manuscript in order to present a clearer presentation. The detail response as below.

Comment 1.

Abstract: it is right, but you have to include a mention to methodology.

Response to Comment 1:

We have added the information about the model of this article as a narrative review (L18). The explanation of the narrative review is available in L94-96 and L100-101.

Comment 2

Theoretical framework: it is correct. Check if some references can be updated. To be prudent, try to update some new references if you find.

Response Comment 2

We have added some new and updated literature to support our paper.

Comment 3

-Method: the principal lack of this paper is Methodology. The question is that there is not the method in a scientific strict sense. You must use a quantitative or qualitative method. I can guess a qualitative indirect method (more or less, a sort of case study messed with content analysis), but it is not clear. Although the main objective of the research is to purpose a prospective activity, a scientific paper needs a method, at least. In this framework, I suggest in-deep interviews or a Delphi.

Response to Comment 3.

As we explained in the introduction response, our paper does not have a specific methodology such as qualitative (interview) quantitative (survey, secondary data, ect.) that is usually available in empirical studies. Therefore, our data collection method is a literature review, that we collect the data/information from previous published papers/articles (research papers, books, reports, and government policies), and analyzed it to formulate a strategy.

Comment 4.

-Results. The absence of method implicates a lack or results.

-Conclusion and discussion: You have to include a more in-depth discussion comparing your findings with the principal authors of the theoretical framework. Conclusion is too much brief. There is a clear disequilibrium between the article in general and conclusion.

Response to Comment 4

Regarding the result and discussion section, we also refer to our response to comment 3. The result of our study is the collected information on ecotourism management in Sumatra, the existing condition, the problems, and the potency (Section 3,4, and 5). We do not provide a discussion section as it is not relevant to a review article.

Round 2

Reviewer 2 Report

Analyzing the submitted article, we found that the authors added certain pieces of text, but without changing the structure of the paper specific to a review article. The article continues to focus on the specific issues of a single area (albeit representative), adding only brief comparisons with other regions. We do not identify the profile of the ecotourist in the area, we do not have relevant statistical data on ecotourism issues. The strategic proposals for the development of ecotourism in Sumatra find their place rather in an official report and less in a scientific article. The conclusions are brief and do not clarify the objectives of a review article.

Author Response

Dear Reviewer 2,

We thank you for your review. 

We've tried to improve as best we can.  We have improved according to the suggestions of the reviewers. We will fix the cohesion and English in our article as directed by the editors of Sustainability.  We've also followed the advice to do language editing on the language editing services available at Sustainability. We hope that this manuscript can be published in the Journal of Sustainability.

Sincerely yours

Authors

Reviewer 4 Report

The authors have improved a little bit some parts, but it remains the principal lack of this paper: Methodology. They accept there is not method in a scientific strict sense, and they mention a narrative method, but it is not enough, at least in the standard international journals.

Author Response

Dear Reviewer 4,

We thank you for your review. 

We've tried to improve as best we can.  We have improved according to the suggestions of the reviewers. We will fix the cohesion and English in our article as directed by the editors of Sustainability.  We've also followed the advice to do language editing on the language editing services available at Sustainability. We hope that this manuscript can be published in the Journal of Sustainability.

Sincerely yours

Authors
